# Fungal Communities in Re-Emerging *Fraxinus excelsior* Sites in Lithuania and Their Antagonistic Potential against *Hymenoscyphus fraxineus*

**DOI:** 10.3390/microorganisms10101940

**Published:** 2022-09-29

**Authors:** Remigijus Bakys, Gintarė Bajerkevičienė, Alfas Pliūra, Adas Marčiulynas, Diana Marčiulynienė, Jūratė Lynikienė, Valeriia Mishcherikova, Audrius Menkis

**Affiliations:** 1Institute of Forestry, Lithuanian Research Centre for Agriculture and Forestry, Liepu˛ Str. 1, LT-53101 Girionys, Kaunas District, Lithuania; 2Department of Forestry, Kaunas Forestry and Environmental Engineering University of Applied Sciences, Liepu˛ Str. 1, LT-53101 Girionys, Kaunas District, Lithuania; 3Department of Forest Mycology and Plant Pathology, Uppsala BioCenter, Swedish University of Agricultural Sciences, P.O. Box 7026, SE-75007 Uppsala, Sweden

**Keywords:** ash dieback, biocontrol, dual-culture assay, fungal communities

## Abstract

Fifty-nine fungal taxa, isolated from re-emerging *Fraxinus excelsior* sites in Lithuania, were in vitro tested against three strains of *Hymenoscyphus fraxineus* on agar media to establish their biocontrol properties. All tested fungi were isolated from leaves and shoots of relatively healthy *Fraxinus excelsior* trees (<30% defoliation), which were affected by ash dieback but their phytosanitary condition has not worsened during the last decade. The inhibition of *H. fraxineus* growth by tested fungal taxa ranged between 16–87%. Occasionally isolated fungal taxa such as *Neonectria coccinea*, *Nothophorma quercina,* and *Phaeosphaeria caricis* were among the most effective fungi inhibiting the growth of *H. fraxineus* cultures. Among the more commonly isolated fungal taxa, *Cladosporium* sp., *Fusarium* sp., *Malassezia* sp., and *Aureobasidium pullulans* showed a strong growth inhibition of *H. fraxineus*.

## 1. Introduction

Over the last two decades, the phenomenon of ash dieback of common ash (*Fraxinus excelsior* L.), caused by the invasive ascomycete *Hymenoscyphus fraxineus*, was reported in many European countries [1,2]. This disease results in high mortality rates and threatens *F. excelsior* species on a continental scale and at the same time threatens the existence of many ash-associated organisms and specific ash-dependent forest ecosystems [3]. In Lithuania, ash dieback emerged in the mid-1990s. Since then, the total area of *F. excelsior* stands in the country due to the impact of this epidemic decreased from 50,800 ha in 1995 to 13,013 ha in 2021, or from 2.7% to 0.6% of all forested area [4,5].

Currently, the ash-dieback epidemic in Lithuania is in its chronic phase, causing stand deterioration at much slower rates [6]. In addition, during the last decade, approx. 2000 ha of *F. excelsior* stands have emerged by natural regeneration [5]. Despite the fact that these stands exhibit ash-dieback symptoms of various extent, they are considered to be in stable phytosanitary conditions, i.e., surveys by Lithuanian National Forest Inventory showed that sanitary conditions remained similar during the last ten years. These stands were not subjected to sanitary fellings and due to the current phytosanitary condition were retained by forest managers. 

The aims of the study were to: (a) investigate mycobiome colonizing leaves and shoots of *F. excelsior* trees in re-emerging stands using pure culture fungal isolations; (b) to test biocontrol properties of obtained fungal isolates against *H. fraxineus* in dual-culture antagonistic assays. 

## 2. Materials and Methods

In total, ten *F. excelsior* stands were sampled in different regions of Lithuania (Table 1, Figure 1A). Fruiting bodies of *H. fraxineus* were detected at these sites (Figure 1B). The study sites were selected based on the following criteria: (a) *F. excelsior* constituted at least 40% of a stand composition; (b) these stands were planted or re-emerged naturally in the period of ongoing ash dieback; (c) stands were of a relatively good and stable phytosanitary condition (defoliation up to 30%), which remained similar during the last ten years.

At each study site, 12 trees of a relatively good phytosanitary condition (defoliation <30%) and situated at least 50 m from each other, were selected for sampling. Sampling was carried out in July 2019 by cutting live twigs with leaves from the lower part of the crown (4–12 m above the ground). In total, 360 samples of leaves and 360 samples of shoots (three samples of leaves and three of shoots per each tree) were collected. Within 24 h after the collection, samples were surface sterilized (brief flaming was used for shoot samples and dipping in 70% ethanol for leaf samples) and placed on 9 cm Petri dishes containing Hagem agar media [7], which was amended with 50 g/L sterilized fresh *F. excelsior* leaves to facilitate the growth of *H. fraxineus*. Plates were stored at room temperature (ca. 21°C) and checked daily for mycelial growth. Emerging fungal mycelium was sub-cultured into new plates to isolate pure cultures. Obtained pure fungal cultures were checked under a light microscope and, according to their morphological structures, were divided into different morphological groups (presumed individual taxa). One representative from each morphological group was subjected to both molecular identification of fungal taxa and antagonistic tests against three *H. fraxineus* strains. Molecular identification of fungal taxa was carried out using sequencing of fungal ITS rRNA as described in Marčiulynas et al. [8]. 

Antagonistic reaction of isolated fungi against *H. fraxineus* strains, which were also isolated during this study, was assessed using dual-culture assays and by evaluating the pattern and growth rate of *H. fraxineus* [9,10]. Petri dishes containing Hagem agar medium were pre-inoculated with an agar plug (5 × 5 mm) of respective *H. fraxineus* strains and left for 5 days at room temperature in the dark. On the sixth day of assay, each Petri dish with a growing colony of *H. fraxineus* was inoculated with an agar plug containing actively growing mycelia of the tested fungus (ca. two-week-old colony was used), which was placed at the distance of 3 cm from the point of *H. fraxineus* inoculation. Controls were established similarly only using *H. fraxineus* strains alone. All tests were established in triplicates. After three weeks, the rate of *H. fraxineus* growth inhibition was assessed using the formula Rk − ((Rh2 − Rh1)/2)/Rk × 100, where Rk is a radial mycelial growth of *H. fraxineus* colony in control plates, Rh2 is a radial mycelial growth of *H. fraxineus* mycelium in dual culture plates, and Rh1 is a radial mycelial growth of *H. fraxineus* mycelium towards the inoculation point of the tested fungus. 

Inhibition rates caused in each in vitro assay were compared using one-way ANOVA and Tukey’s test for multiple comparisons between all pairs of means. The statistical analysis was performed using Minitab v.19 (Minitab Inc., State College, PA, USA) statistical software package.

## 3. Results and Discussion

Isolation and molecular identification showed the presence of 59 fungal taxa (Table 2). 

Out of these, 56 fungal taxa were identified at least to the genus level, while three taxa remained unidentified. Most frequently isolated fungi from shoots were *Alternaria alternata*, *Apiospora montagnei*, *Gibberella avenacea*, and *Phoma* sp. (Table 2). Most frequently isolated fungi from leaves were *A. alternata*, *Aureobasidium pullulans*, *Apiospora montagnei*, and *G. avenacea*. The fungal taxa detected in shoots and leaves of *F. excelsior* were similar to those reported in numerous other culture-based studies [11,12,13,14]. However, high-throughput sequencing of the shoot and leave samples collected in the present study revealed a very contrasting fungal community structure [15]. Evidently, each of the aforementioned fungal detection methods has its advantages and shortcomings, suggesting that the use of both methods can provide valuable complementary information.

Dual culture assays revealed the presence of three types of fungal interactions: A—deadlock; B—antibiosis; and C—replacement (Figure 2).

During dual culture assays, the B type of mycelial interactions was the most common output (41.1%), while A and C types were observed less frequently (30.3% and 28.6%, respectively) (Figure 3). 

In comparison to self-inhibition of *H. fraxineus* in dual-culture assays, thirty-six tested fungal taxa significantly suppressed the growth of *H. fraxineus*, while eight of the tested taxa significantly enhanced the growth of *H. fraxineus* (*p* < 0.05, Figure 3). Among the fungi most effectively limiting the growth of *H. fraxineus* (Figure 3), there were the pathogenic ascomycete *Nothophorma quercina*, which is known as a fungus causing shoot cankers and leaf spots [16], and *Phaeosphaeria caricis* (inhibition factor 87% and 82%, respectively). The other fungi effectively inhibiting the growth of *H. fraxineus* were *Neonectria coccinea*, *Lophiostoma corticola*, *Cytospora pruinosa*, *Penicillium flavigenum*, *Cladosporium sp.*, and *Fusarium solani* (inhibition factor between 75% and 78%, Figure 3). The rate of growth inhibition of *H. fraxineus* for remaining fungal taxa ranged between 16% and 74%. Among the fungal taxa frequently detected in leaves and shoots of *F. excelsior*, *Malassezia* sp. (inhibition factor 74%) and *A. pullulans* (71%) showed a strong antagonistic reaction against *H. fraxineus*. In comparison, the interspecific dual-culture test between the two different *H. fraxineus* strains showed that the inhibition factor was 57%. *Cladosporium* sp. (inhibition factor 76%) was relatively infrequently isolated in the present study (Table 2), but it was among the dominating fungi in a high-throughput sequencing study [15], therefore this particular fungal taxon can be regarded as frequent in nature. 

We conclude that the results of the present study provided valuable information on the potential suitability of different fungal taxa for the biocontrol of *H. fraxineus*. Among all fungal taxa tested only *Cladosporium* sp. and *A. pullulans* can be considered as frequent in nature and effective growth inhibitors of *H. fraxineus*. Several species within the genus *Cladosporium*, which is the common cosmopolitan endophytic genus, were previously shown in a number of studies as potential agents in protecting plants against different biotic and abiotic stresses and as enhancers of plant growth [17,18,19]. *Aureobasidium pullullans* was shown to be able to inhibit post-harvest pathogens of fruits [20]. Results of dual-culture assays in our study are in agreement with the results of other similar studies [10,21] where a significant effect on the growth inhibition of *H. fraxineus* by these two fungal taxa was also reported. In support, a negative correlation between *H. fraxineus* and *Cladosporium* sp. was also found in shoots of *F. excelsior* [15]. *Cladosporium* sp. in our tests was not only among the fungi showing the strongest reaction of inhibition of *H. fraxineus*, but it also was able to show the overgrowth of *H. fraxineus*, suggesting the potential antagonism in nature [9]. On the other hand, *A. pullulans* and *Cladosporium* sp. were frequently reported as fungi associated with *F. excelsior* [13,22,23], indicating that *H. fraxineus* can possibly co-exist with these ascomycetes in vivo or it is able to avoid the direct confrontation due to seasonal succession of these fungi on tissues of *F. excelsior* [13]. It should also be noted that fungal taxa, which showed the strongest reaction of growth inhibition of *H. fraxineus* in dual-culture assays, were fast-growing fungi. The possibility should not be excluded that this property facilitated the overgrowth of *H. fraxineus*, which generally exhibits slower growth rates. Therefore, it is possible that the inhibition of *H. fraxineus* in tissues of *F. excelsior* depends on the composition of the entire fungal community rather than on a few individual fungal taxa. Biocontrol tests using selected fungal taxa under in vivo conditions could provide new valuable insights on this important research topic.

## Figures and Tables

**Figure 1 microorganisms-10-01940-f001:**
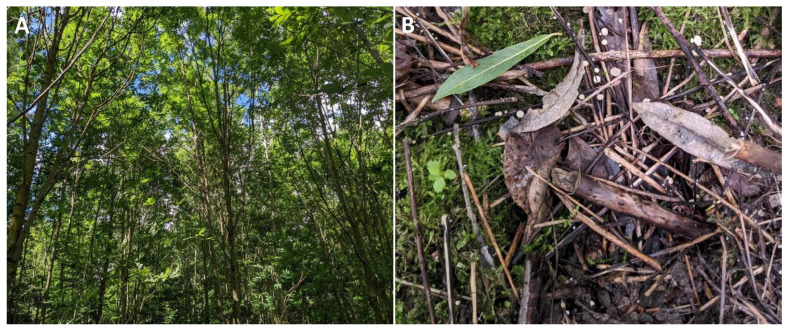
(**A**) Naturally re-emerged *Fraxinus excelsior* stand (site U2); (**B**) *Hymenoscyphus fraxineus* fruiting bodies on ash petioles at the U2 site.

**Figure 2 microorganisms-10-01940-f002:**
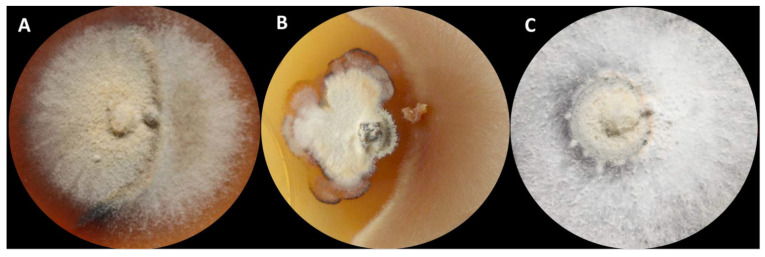
The outcome of dual culture screening assay between *Hymenoscyphus fraxineus* (on the left) and different tested fungi (on the right). (**A**) Both fungi meet and form a barrage zone–deadlock reaction; (**B**) The inhibition zone is formed between *H. fraxineus* and the tested fungus–antibiosis reaction; (**C**) The tested fungus shows the overgrowth of *H. fraxineus*–replacement reaction.

**Figure 3 microorganisms-10-01940-f003:**
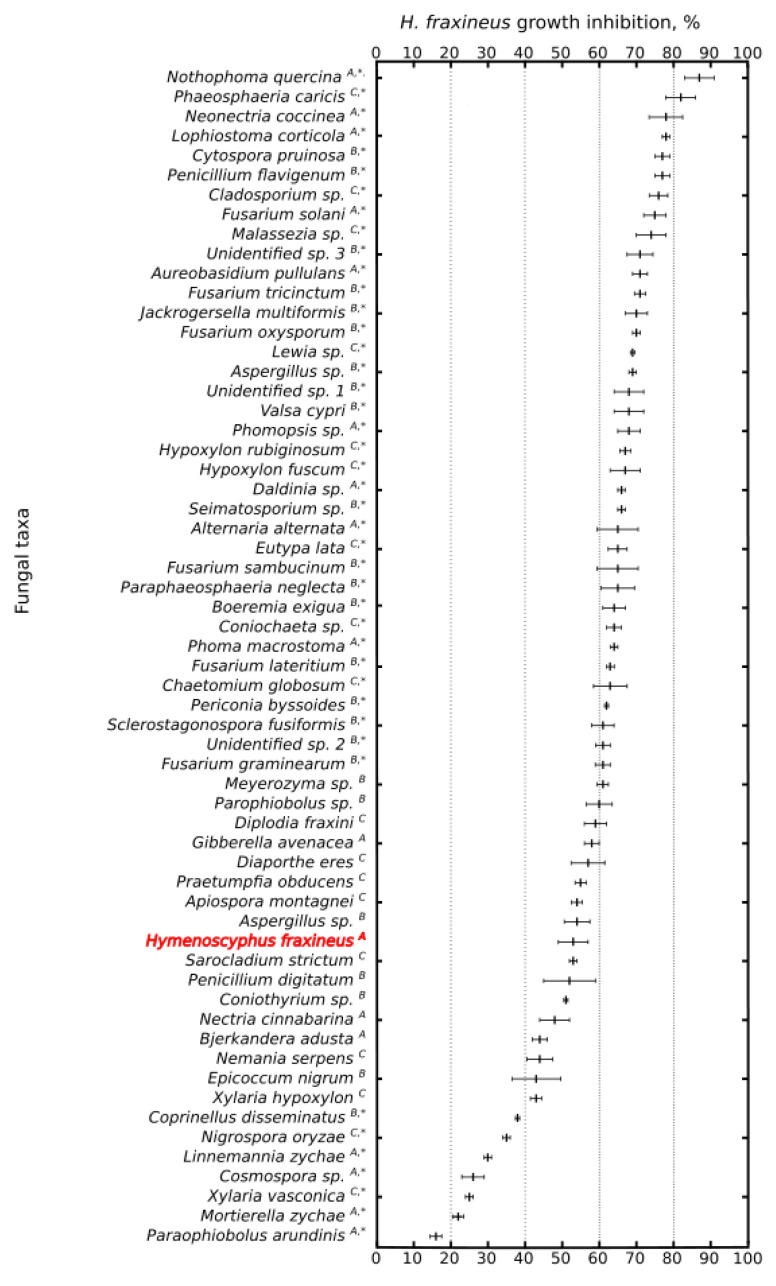
The effect of grown inhibition of *Hymenoscyphus fraxineus* by fungal taxa isolated from shoots and leaves of *Fraxinus excelsior*. Next to the species names, A denotes A type (deadlock) of mycelial interaction; B—B type (antibiosis); and C—C type (replacement). The symbol * denotes significantly higher or significantly lower growth inhibition of *H. fraxineus* compared to *H. fraxineus* self-inhibition (marked in red). The output of interaction is shown as the average value of inhibition factor ± SD. X-axis represents inhibition rate (%).

**Table 1 microorganisms-10-01940-t001:** *Fraxinus excelsior* sites in Lithuania investigated in the present study.

Site	Latitude, N	Longitude, E	Tree Age, y	Tree Species Composition, % *	Mean Annual Temperature, °C	Mean Annual Precipitation, mm	Stand Origin **
U1 (Žeimelis)	56°16′37.1″	24°02′33.0″	10–25	70F,10P,10B,10U	6.3	620	S
U2 (Kėdainiai)	55°13′35.4″	23°57′58.5″	15–20	90F,10P	6.7	600	S
U3 (Šakiai)	55°01′49.8″	23°00′20.2″	15–20	100F	7.0	650	S
U4 (Telšiai)	56°03′43.4″	22°25′08.3″	17	70F,30Q	6.3	800	P
U5 (Šakiai)	55°00′50.2″	23°04′00.9″	15–25	70F,10P,10B,10S	7.0	650	S
U6 (Dubrava)	54°44′20.6″	23°47′38.8″	17	100F	6.8	650	P
U7 (Telšiai)	56°03′33.8″	22°25′37.6″	17	70F,30Q	6.3	800	P
U8 (Pakruojis)	56°00′38.4″	23°55′48.5″	15–20	90F,10P	6.5	600	S
U9 (Jonava)	55°09′11.1″	24°06′56.8″	7–10	70F,30Q	6.7	620	S
U10 (Dubrava)	54°45′07.4″	23°47′53.5″	15–17	80F,10P,10B	6.8	650	S

* F—*Fraxinus excelsior*, P—*Populus tremula*, Q—*Quercus robur*, B—*Betula pendula*, U—*Ulmus sp.*, S—*Picea abies*, ** S—self-regenerated, P—progeny trial.

**Table 2 microorganisms-10-01940-t002:** Frequencies of fungi isolated from shoots and leaves of F. excelsior.

Fungal Taxa	No. of Pure Cultures/%	
	Shoots	Leaves
*Alternaria alternata*	87/10.9	71/22.5
*Apiospora montagnei*	65/8.2	44/13.9
*Aspergillus* sp.	2/0.3	-/-
*Aspergillus* sp.	6/0.8	-/-
*Aureobasidium pullulans*	6/0.8	43/13.6
*Bjerkandera adusta*	12/1.5	-/-
*Boeremia exigua*	14/1.8	12/3.8
*Chaetomium globosum*	24/3.0	-/-
*Cytospora pruinosa*	2/0.3	-/-
*Cladosporium allicinum*	6/0.8	7/2.2
*Coniochaeta* sp.	4/0.5	-/-
*Coniothyrium* sp.	3/0.4	-/-
*Coprinellus disseminatus*	9/1.1	-/-
*Cosmospora* sp.	1/0.1	3/0.9
*Daldinia* sp.	2/0.3	-/-
*Diaporthe eres*	21/2.6	4/1.2
*Diplodia fraxini*	11/11.4	3/0.9
*Epicoccum nigrum*	49/6.2	14/4.4
*Eutypa lata*	21/2.6	12/3.8
*Fusarium graminearum*	3/0.4	-/-
*Fusarium lateritium*	7/0.9	4/1.2
*Fusarium oxysporum*	3/0.4	2/0.6
*Fusarium sambucinum*	2/0.3	-/-
*Fusarium solani*	1/0.1	1/0.3
*Fusarium tricinctum*	2/0.3	2/0.6
*Gibberella avenacea*	54/6.8	24/7.6
*Hymenoscyphus fraxineus*	1/0.1	-/-
*Hypoxylon fuscum*	19/2.4	3/0.9
*Hypoxylon rubiginosum*	13/1.6	5/1.5
*Jackrogersella multiformis*	2/0.3	-/-
*Lewia* sp.	16/2.0	-/-
*Linnemannia zychae*	1/0.1	-/-
*Lophiostoma corticola*	4/0.5	-/-
*Malassezia* sp.	52/6.5	7/2.2
*Meyerozyma* sp.	3/0.4	-/-
*Mortierella zychae*	6/0.8	14/4.3
*Nectria cinnabarina*	3/0.4	-/-
*Nemania serpens*	7/0.9	2/0.6
*Neonectria coccinea*	5/0.6	-/-
*Nigrospora oryzae*	12/1.5	3/0.9
*Nothophoma quercina*	1/0.1	-/-
*Paraophiobolus arundinis*	2/0.3	-/-
*Paraphaeosphaeria neglecta*	3/0.4	4/1.2
*Penicillium digitatum*	20/2.5	-/-
*Penicillium flavigenum*	6/0.8	-/-
*Periconia byssoides*	17/2.1	7/2.2
*Phaeosphaeria caricis*	3/0.4	4/1.2
*Phoma macrostoma*	67/8.4	11/3.4
*Phomopsis* sp.	15/1.9	-/-
*Praetumpfia obducens*	3/0.4	-/-
*Sarocladium strictum*	44/5.5	7/2.2
*Sclerostagonospora fusiformis*	3/0.4	-/-
*Seimatosporium* sp.	2/0.3	-/-
*Valsa cypri*	15/0.9	-/-
*Xylaria hypoxylon*	2/0.3	2/0.6
*Xylaria vasconica*	7/0.9	1/0.3
Unidentified sp.	5/0.6	-/-
Unidentified sp.	1/0.1	-/-
Unidentified sp.	2/0.3	-/-
Total	779/100	316/100

## Data Availability

The data is available upon request from the corresponding author.

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
