# Peer review of "Fungal Communities in Re-Emerging Fraxinus excelsior Sites in Lithuania and Their Antagonistic Potential against Hymenoscyphus fraxineus"

_microorganisms, 2022, doi:10.3390/microorganisms10101940_

Round 1
Reviewer 1 Report
Regarding the manuscript by Bakys et al., "Fungal Communities in Re-emerging Fraxinus excelsior Sites in Lithuania and Their Antagonistic Potential Against Hymenoscyphus fraxineus" (ref. microorganisms-1928761) submitted for consideration by the Journal Microorganisms.
The manuscript deals with the screening of culturable fungal species from F. excelsior present in re-emerging Lithuanian sites. Moreover, the authors also conducted inhibition studies against H. fraxineus with the retrieved fungal isolates.
In my opinion the topic is interesting, the manuscript is well written and the methods employed suited. This manuscript does not have any particular major flaws, but could be improved with the addition of some photographic data and a deeper discussion of the results obtained.
Moreover, I believe that a few additional aspects should be addressed before this manuscript can be considered ready for publication:
· The authors state that they have conducted one-way ANOVA and Tukey tests. However, in the current manuscript these results are not properly presented or discussed. Please solve this issue.
· Line 37 “stable phytosanitary condition”. Could the authors provide more details reggarding this stability? Also in Line 47 “and stable phytosanitary condition (defoliation up to 30%)”, is this some national/international criteria or pherhaps it is referenced in other works? Please provide some additional details.
· Lines 38 to 40. The english language of this sentence can be improved.
· Materials and Methods. Could the authors provide some photos of the sampling sites? In addition, while the geographical locations are given, if possible please also add the sampling locations names to Table 1.
· Line 61: “outgrow” please replace by other word.
· Lines 64 to 67. Please provide more details regarding the molecular analysis. In the current state is impossible to have even a small idea of what was conducted.
· Results and Discussion. Please provide some photos of the antagonistic dual-culture assays conducted. Moreover, Lines 108 to 111, Please provide some comparative photos of these results.
· Lines 134 to 136. Please discuss in more detail comparing with the available literature.
Author Response
Dear Reviewer, please find our point-to point response below:
- The authors state that they have conducted one-way ANOVA and Tukey tests. However, in the current manuscript these results are not properly presented or discussed. Please solve this issue.
Response: significant differences as compared to controls were indicated in Fig. 3 and discussed in the manuscript.
- Line 37 “stable phytosanitary condition”. Could the authors provide more details reggarding this stability? Also in Line 47 “and stable phytosanitary condition (defoliation up to 30%)”, is this some national/international criteria or pherhaps it is referenced in other works? Please provide some additional details.
Response: additional details were provided
- Lines 38 to 40. The english language of this sentence can be improved.
Response: the structure of the sentence was changed.
- Materials and Methods. Could the authors provide some photos of the sampling sites? In addition, while the geographical locations are given, if possible please also add the sampling locations names to Table 1.
Response: photos showing one of the sampling sites and the evidence of H. fraxineus were included. The names of sampling locations were added to Table 1.
- Line 61: “outgrow” please replace by other word.
Response: replaced
- Lines 64 to 67. Please provide more details regarding the molecular analysis. In the current state is impossible to have even a small idea of what was conducted.
Response: additional information was included
- Results and Discussion. Please provide some photos of the antagonistic dual-culture assays conducted. Moreover, Lines 108 to 111, Please provide some comparative photos of these results.
Response: photos showing different outcomes of dual culture assays were included
- Lines 134 to 136. Please discuss in more detail comparing with the available literature.
Response: the discussion in indicated place was more thoroughly explicated

Reviewer 2 Report
Dear Authors,
Most of my suggestions are editorial and can be found in the attached file.
Best wish

Author Response
Dear Reviewer, please find our response to your comments as follow:
Multiple comments and suggestions found in a PDF file were carefully considered and changes were made accordingly.
- Please clarify sentence “which were ash dieback affected but with a stable phytosanitary condition”
Response: additional information was included to Abstract and Introduction sections of the manuscript.
This part is too short (lines 30-31), you should give more detail about Hymenoscyphus fraxineus and Fraxinus excelsior.
Response: Introduction was expanded as suggested.
Mention how old of your fungi strains and what size…. Where did you get this pathogen?
Response: additional information was added to the Materials and Methods section.
Please provide some images to show the inhibition
Response: photos showing different outcomes of dual culture assays were included.
Please clarify based on what? ITS gene?
Response: additional information sequencing ITS rRNA was included to Materials and Methods section.
Even unpublished, you need to cite the paper here and add a reference in the list and mention it in press
Response: corrected.

Round 2
Reviewer 1 Report
The majority of my comments were addressed.
Some final considerations:
Lines 16 and 17 …were in vitro tested against…
Line 32 This disease results
Line 45 to be in stable
Line 47 sanitary
Line 57 there is a period missing
Author Response
Reviewer #1
Dear Reviewer,
Please find attached the revised version of the manuscript. All comments were considered, and changes were made accordingly in the manuscript using MS Word Trach-changes.
Thank you for your consideration.
Sincerely,
Remigijus Bakys
Point-by-point responses:
Lines 16 and 17 …were in vitro tested against…
Response: corrected.
Line 32 This disease results
Response: corrected.
Line 45 to be in stable
Response: corrected.
Line 47 sanitary
Response: corrected.
Line 57 there is a period missing
Response: corrected.

Reviewer 2 Report
Dear Authors,
I reviewed this manuscript before, and most of my suggestions are considered. However, there are a few issues I would like to be addressed in this version.
Best wishes

Author Response
Reviewer #2
Dear Reviewer,
Please find attached the revised version of the manuscript. All comments were considered, and changes were made accordingly in the manuscript using MS Word Trach-changes.
Thank you for your consideration.
Sincerely,
Remigijus Bakys
Point-by-point responses:
Table 2. Several still used commas, not dots
Response: corrected.
Line 137-138. Multiple corrections
Response: corrected.
Line 235-236 Simple word.
Response: corrected.
